# Comparison of Bone Regeneration in Different Forms of Bovine Bone Scaffolds with Recombinant Human Bone Morphogenetic Protein-2

**DOI:** 10.3390/ijms222011121

**Published:** 2021-10-15

**Authors:** Hyun Seok, Hee-Youl Kim, Dong-Cheol Kang, Jung-Ho Park, Jong Hoon Park

**Affiliations:** 1Department of Oral and Maxillofacial Surgery, School of Dentistry, Jeonbuk National University, Jeonju 54896, Korea; hiar1212@naver.com (H.-Y.K.); rkdehdcjf12@naver.com (D.-C.K.); Junghopark1106@gmail.com (J.-H.P.); themobius8@naver.com (J.H.P.); 2Research Institute of Clinical Medicine of Jeonbuk National University-Biomedical Research Institute of Jeonbuk National University Hospital, Jeonju 54896, Korea

**Keywords:** bone morphogenetic protein, bovine bone, bone regeneration, adipose tissue

## Abstract

The aim of this study was to compare the bone regeneration ability of particle and block bones, acting as bone scaffolds, with recombinant human bone morphogenetic protein (rhBMP)-2 and evaluate them as rhBMP-2 carriers. Demineralized bovine bone particles, blocks, and rhBMP-2 were grafted into the subperiosteal space of a rat calvarial bone, and the rats were randomly divided into four groups: particle, block, P (particle)+BMP, and B (block)+BMP groups. The bone volume of the B+BMP group was significantly higher than that of the other groups (*p* < 0.00), with no significant difference in bone mineral density. The average adipose tissue volume of the B+BMP group was higher than that of the P+BMP group, although the difference was not significant. Adipose tissue formation was observed in the rhBMP-2 application group. Histologically, the particle and B+BMP groups showed higher formation of a new bone. However, adipose tissue and void spaces were also formed, especially in the B+BMP group. Hence, despite the formation of a large central void space, rhBMP-2 could be effectively used with block bone scaffolds and showed excellent new bone formation. Further studies are required to evaluate the changes in adipose tissue.

## 1. Introduction

A sufficient width and height of the alveolar bone are required for the installation of dental implants and prosthetic rehabilitation in edentulous patients [1]. Advanced alveolar bone resorption can be attributed to periodontal disease, trauma, pathological conditions, and infectious diseases. Numerous surgical techniques have been introduced for the reconstruction of alveolar bone defects [2]. In recent years, allogenous, xenogenic, and synthetic bones have been developed as bone substitute materials and have contributed to new bone formation. Dental implants can be placed in the alveolar bone defect area [3,4,5]. Autogenous bone is considered the gold standard bone graft material because of its osteogenic, osteoinductive, and osteoconductive abilities. However, donor site morbidity observed upon bone harvest is a significant drawback of this technique [6]. Demineralized bovine bone is popularly used as a xenograft, and although it only exhibits osteoconductive properties, it demonstrates a favorable outcome for new bone formation in the grafted area [7,8].

Growth factors that allow the migration and differentiation of osteogenic cells enhance new bone formation and increase the quality and quantity of newly formed bones [9]. Bone morphogenetic proteins (BMPs) were first discovered by Dr. Marshall Urist, an orthopedic surgeon [10]. BMPs, typically used as growth factors, are members of the transforming growth factor-beta (TGF-β) superfamily and play a role in skeletal organogenesis [11]. BMPs released by platelets and osteoprogenitor cells contribute to osteoblast proliferation, differentiation, and bone formation [12]. BMPs are soluble and low molecular weight transmembrane glycoproteins that bind to and activate a transmembrane receptor complex. The formation of the ligand/receptor complex activates intracellular signaling, resulting in the activation of the transcription of target genes [13]. Twenty types of BMPs have been identified, and only BMP-2, -4, -6, -7, and -9 have been known to exhibit osteogenic properties [11,14]. Advances in molecular biology have allowed the sequencing and cloning of BMP, and human complementary DNA clones of BMP have been used to produce recombinant BMPs in mammalian and non-mammalian cells [11,15]. Recombinant human BMP (rhBMP)-2 and 7 were first approved for orthopedic applications, and rhBMP-2 was later approved for craniofacial application [9]. Currently, rhBMP-2, a growth factor, is used as a bone substitute for alveolar bone reconstruction and dental implant installation [9].

Several biomaterials and bone scaffolds have been introduced and evaluated as rhBMP-2 carriers. An absorbable collagen sponge (ACS) was first used as an rhBMP-2 carrier in maxillary sinus augmentation, and it seems to be a safe and effective alternative for new bone formation [16]. Allogenic bone, bovine bone, hydroxyapatite, and biphasic calcium phosphate (BCP) are also used with rhBMP-2 in maxillary sinus augmentation, which enhances new bone formation compared to bone substitutes alone [17]. In alveolar ridge augmentation, rhBMP-2 with a particulate bone scaffold provides favorable bone gain, allowing the placement of dental implants [4]. Various combinations of rhBMP-2 and scaffolds have been used for alveolar bone reconstruction. Although the development of postoperative edema is a significant concern when using a relatively high concentration of rhBMP-2, it shows favorable results in bone formation and efficacy as an osteogenic agent [18].

Various forms or types of bone scaffolds, such as particles, putty, and block bovine bone, have been manufactured and used in the clinical field [5,19]. The clinical application of each bone substitute can vary according to defect size, location, and extent. It is also affected by the purpose of the bone graft, the morphology of the recipient bed, and surgeon preference [20]. rhBMP-2 is combined with various forms of scaffolds, including collagen sponges, bone matrix gels, blocks, and particle bones [3,21,22]. Although it shows favorable results for bone regeneration, studies examining the optimal or best form of bone scaffolds, such as rhBMP-2 carriers, have not been performed. The several forms of bone scaffolds available need to be evaluated as rhBMP-2 carriers and compared with each other. Therefore, in this study, we aimed to compare the bone regeneration ability in different forms of bovine bone scaffolds with rhBMP-2 and evaluate the efficacy of different forms of bone scaffolds as rhBMP-2 carriers.

## 2. Results

### 2.1. Morphology of Bone Scaffold by Field Emission Scanning Electron Microscope (FE-SEM)

The morphology of the bone scaffolds, which were particle and block bovine bone, are presented in Figure 1. The particle bovine bone was differently sized and irregularly shaped. It had a highly rough and peaked surface at high magnification (Figure 1b). The block bovine bone was a solid bone substitute mass with an irregular surface. Various sized particles and chips were distributed on the surface of the block bone (Figure 1d).

### 2.2. Micro-Computed Tomography (μ-CT) Analysis

Three-dimensional (3D) reconstruction images of μ-CT images are shown in Figure 2. The grafted bone scaffold was observed on the surface of the calvarial bone in all groups 6 weeks after surgery. The particle and block groups retained their original appearance as bone scaffolds without significant changes. The P (particle)+BMP and B (block)+BMP groups showed new bone formation and mineralization on the outer surface of the grafted bone scaffold. Newly formed bone was observed inside the bone scaffold in the block, P+BMP, and B+BMP groups on the section image. 

The results of the μ-CT analysis for each group are shown in Figure 3. The average bone volumes (BVs) of the particle, block, P+BMP, and B+BMP groups were 60.42 ± 20.35, 94.46 ± 20.05, 129.87 ± 21.92, and 213.76 ± 70.45 mm^3^, respectively, at 6 weeks after the operation (Figure 3a). Significant differences were observed between the BVs of B+BMP and the other groups (*p* < 0.000). The BV of the B+BMP group was significantly higher than that of the particle (*p* = 0.000), block (*p* = 0.002), and P+BMP groups (*p* = 0.007). The BV of the P+BMP group was significantly higher than that of the particle group (*p* = 0.04). The average bone mineral densities (BMDs) of the particle, block, P+BMP, and B+BMP groups were 748.17 ± 37.56, 729.14 ± 40.35, 768.08 ± 78.52, and 681.17 ± 103.36 mg/cc, respectively. No significant difference was observed in BMD among the four groups (*p* > 0.05) (Figure 3b). 

The average trabecular thickness (TbTh) values of the particle, block, P+BMP, and B+BMP groups were 0.46 ± 0.03, 0.49 ± 0.01, 0.50 ± 0.13, and 0.58 ± 0.08 mm, respectively (Figure 3c). The average TbTh of the B+BMP group was higher than that of the other groups. No significant differences were observed between the values among all the groups (*p* > 0.05). The average trabecular spacing (TbSp) values of the particle, block, P+BMP, and B+BMP groups were 0.20, 0.15 ± 0.01, 0.20 ± 0.07, and 0.24 ± 0.08 mm, respectively (Figure 3d). The average TbSp of the B+BMP group was higher than that of the other groups. No significant differences were observed between the values among all the groups (*p* > 0.05).

### 2.3. Adipose Tissue Formation in the P+BMP and B+BMP Groups

Void and vacant spaces are shown in the sectional image of the μ-CT analysis (Figure 4a). The void space was assumed to be formed by the formation of adipose tissue inside the new bone. This space was observed in the newly formed bone and bone scaffolds in the P+BMP and B+BMP groups. In particular, a large adipose tissue was observed in the B+BMP group, which created a vacant space in the center area of the bone scaffold. The volume of adipose tissue was measured and is presented in Figure 4b. The average adipose tissue volume (ATV) values of the P+BMP and B+BMP groups were 25.87 ± 16.03 and 65.81 ± 43.65 mm^3^. No significant differences were observed in ATV among all of the groups (*p* = 0.063).

### 2.4. New Bone and Adipose Tissue Formation Observed after Histological Evaluation

Histological images of each group are shown in Figure 5. The grafted bone scaffold was observed on the calvarial bone surface in the particle and block groups. Connective and fibrotic tissues occupied the space between the grafted bone scaffold, and there was no sign of new bone formation in either group (Figure 5a,b). In the P+BMP and B+BMP groups, new bone formation was observed in the bone scaffold with bone maturation and mineralization. Adipose tissue formation was also observed inside the newly formed bone in these groups. In the P+BMP group, the adipose tissue formed a void space and was dispersed and distributed throughout the particle bone scaffold (Figure 5c). In contrast, in the B+BMP group, a large adipose tissue was observed in the center of the block bone scaffold and was surrounded by new bone (Figure 5d).

### 2.5. Osteogenic Marker Expression in Immunohistochemistry (IHC)

The results of bone sialoprotein (BSP) and osteocalcin expression observed by IHC are presented in Figure 6. The particle and block groups showed low expression of BSP in the connective tissue inside the bone scaffold. The P+BMP and B+BMP groups showed high expression of BSP in the newly formed bone and osteocytes inside the new bone matrix (Figure 6). Low expression of osteocalcin was observed in the particle and block groups, and high expression of osteocalcin was observed in the new bone matrix of the P+BMP group. Osteocalcin expression was also observed in the new bone adjacent to the bone scaffold in the B+BMP group (Figure 6).

## 3. Discussion

The block type of the bone scaffold can be used in onlay bone grafts for gains in the length and width of the alveolar bone. Autologous block bone harvested from the mandibular ramus and symphysis has been frequently used for vertical bone augmentation in the alveolar bone resorption area [6]. It offers the advantages of favorable bone regeneration that provides osteogenic cells, osteoinductive growth factors, and bone matrix scaffolds. However, it also has disadvantages, such as the morbidity associated with the harvested area, increased operation time, and limited harvest amount. Bovine block bone can be a candidate as an alternative to autogenous block bone for vertical bone augmentation [23]. Block bone scaffolds have three-dimensional structures, provide space for osteogenic cell migration, and maintain the volume for new bone formation [23]. In our study, the block bone with osteoconductive activity was used with osteoinductive growth factors, such as rhBMP-2, to enhance the efficacy of bone regeneration and the block bone was evaluated as a BMP carrier. Under in vivo conditions, bovine block bone was successfully grafted with rhBMP-2 for vertical bone augmentation [24]. The allogenic block bone scaffold with rhBMP-2 was used for horizontal augmentation and to obtain a sufficient width of the alveolus in the anterior maxilla [19].

Urist discovered that the demineralized bone matrix induces cartilage, bone, and marrow formation when implanted in the skin or muscle tissue of rodents [10]. The bone-inducing factor responsible for new bone formation at the ectopic site was observed in the demineralized bone matrix and was later named “bone morphogenetic proteins” [10,25]. Currently, more than 20 types of BMPs have been identified as members of the TGF-β superfamily, and several of them exhibit the ability to induce bone formation. BMPs play a critical role in the embryonic development of skeletal and non-skeletal tissues and are related to the formation of various organs, such as bone, cartilage, muscle, and vessels [26,27]. BMPs induce the proliferation or differentiation of osteoblasts from mesenchymal stem cells by activating osteogenic transcription factors. Several BMPs play an important role in adipogenesis and induce adipogenic differentiation by activating peroxisome proliferator-activated receptor gamma (PRARγ) signaling [28,29]. Several BMPs exhibit osteogenic properties, among which rhBMP-2 and rhBMP-7 are being pursued for the reconstruction or repair of craniofacial regions [9]. rhBMP-2 has recently been approved for oral and maxillofacial indications and is now widely used in the clinical field for alveolar bone augmentation and sinus lifting for implant installation [30]. 

In this study, we grafted the particle or block bone scaffold on the subperiosteal space of the calvarial bone, similar to onlay bone grafting without fixation, and compared the bone regeneration efficacy in the different forms of the bovine bone scaffold with and without rhBMP-2. The BV of the particle group was 60.42 ± 20.35 mm^3^, and that of the block group was 94.46 ± 20.05 mm^3^. rhBMP-2 was used as an osteoinductive factor with an osteoconductive scaffold of the particle and block bones as BMP carriers, and it showed considerable new bone formation. rhBMP-2 contributed to and enhanced new bone formation, especially when used with block bone scaffolds. The block bone has a rigid three-dimensional structure and provides space for the migration of osteogenic cells and new bone formation. The higher values of BV and TbTh are considered new bone formation and new bone regeneration [31,32]. The results of BV and ThTb indicated that the block bone was a more suitable scaffold than the particle bone as a BMP carrier.

BMPs affect skeletal development and organogenesis, which regulate the differentiation of various tissue organ cells, including osteoblasts, chondrocytes, tenocytes, and myocytes, forming mesenchymal stem cells [26]. BMPs induce adipogenesis by stimulating the differentiation of adipocytes from mesenchymal progenitor cells, and BMP-2, 4, and 7 have been shown to promote adipogenesis [33,34,35]. Adipose tissue formation is known to be one of the clinical side effects of BMP-2 when used for bone formation. The well-known clinical side effects of BMP-2 are osteoclast activation, adipogenesis, and the inflammatory reaction that leads to postoperative swelling [29]. Adipose tissue formation leads to a void space in the newly formed bone and affects the quality of the new bone [36]. When a high concentration of rhBMP-2 is applied for bone regeneration, void space and cyst-like bone formation can be observed [37,38]. In an in vitro study, the osteogenic and adipogenic potentials of human alveolar bone-derived stromal cells were enhanced by rhBMP-2 in a dose and time-dependent manner [39]. Cyst-like and void spaces comprising adipose tissue are clinically observed in the new bone when rhBMP-2 is used as a bone substitute for maxillary sinus augmentation [40,41].

Histological evaluation of the P+BMP group showed the formation of adipose tissue in the newly formed bone, which was scattered and separated (Figure 5). This histology shows a similar pattern to that observed in a previous case report that showed a histological evaluation of rhBMP-2 mediated sinus graft augmentation [40]. This adipose tissue created vacant and void spaces in the newly formed bone and affected the overall bone quality and BMD despite the increase in BV. In the B+BMP group, the adipose tissue created a large vacant space in the center of the block bone, and a new bone surrounded this space. The total BV of the B+BMP group was significantly higher than that of the other groups (*p* < 0.000); however, the BMD of the B+BMP group was lower than that of the other groups. Higher formation of new bone and adipose tissue was observed in the B+BMP group. The average ATV of the B+BMP group was higher than that of the P+BMP group. It also affected the low bone quality, and BMD observed in the B+BMP group. BV and ATV were higher in the B+BMP group, indicating that the activation of osteogenesis and adipogenesis induced by rhBMP-2 was dominant in the B+BMP group.

In our previous study, 50 µg of rhBMP-2 was used with bovine particle bone, and the BV of that group was 134.88 ± 15.24 mm^3^ [36]. In this study, we used 30 µg of rhBMP-2 with bovine block bone in the B+BMP group. The BV of the B+BMP group was 213.76 ± 70.45 mm^3^, and it showed more BV than that of the group with particle bone with 50 µg of rhBMP-2. The block type of bone scaffold is more rigid and stable than other forms of scaffolds. It may be better to contain and stabilize rhBMP-2 in the scaffold and affect its release. The mechanism of the formation of adipose tissue in the central area of the block bone cannot be determined, suggesting that it may be related to the time of adipocyte differentiation and adipose tissue formation in the bone scaffold. A recent study showed that the void space attributed to the adipogenic activity of rhBMP-2 after maxillary sinus augmentation can disappear during a long-term examination. The volume of the new bone and void space was measured using computed tomography, and the void space volume was significantly reduced after twenty-four months of maxillary sinus augmentation. This clinical report concluded that osteogenesis could progress in the void space, which is filled with new bone and, consequently, void space can disappear clinically [41]. This study was performed using CT analysis, and histological analysis is required for the evaluation of osteogenesis and changes in void space over a long-term period. In our study, we compared new bone regeneration at 6 weeks after surgery, which is a limitation of our study. Further study would be required in different observations or long-term periods after surgery to evaluate the change in the void space and new bone formation.

Although adipose tissue was formed by the adipogenic activity of rhBMP-2, the μ-CT analysis showed significant bone enhancement, including high BV and TbTh, in the rhBMP-2 application groups. Histologically, a higher mature bone formation was observed in the P+BMP and B+BMP groups between the grafted material than that observed with the particle and block groups. Mature and mineralized new bone was observed in the P+BMP and B+BMP groups, which surrounded the vacant space formed by adipose tissue in the B+BMP group (Figure 3). Osteogenic markers were highly expressed in the rhBMP-2 group. BSP is a non-collagenous glycoprotein that is abundantly found in mineralized connective tissues, such as bone, dentin, cementum, and cartilage, and has a role in biomineralization [42]. Osteocalcin is secreted by osteoblasts and is often used as a marker of osteoblast differentiation and bone formation processes [43]. BSP and osteocalcin, as osteogenic markers, were not expressed in the fibrotic tissue of the particle and block groups. However, high expression of BSP was observed in the newly formed bone in the rhBMP-2 application group. Osteocalcin expression was observed in the new bone matrix in the P+BMP group and the new bone adjacent to the grafted material in the B+BMP group (Figure 6). This histological result was consistent with that observed with the μ-CT analysis. Hence, rhBMP-2 showed osteoinductive activity and enhanced new bone formation, especially when used with a block bone scaffold in the subperiosteal bone graft.

In our study, we used only 30 µg of rhBMP-2 to compare the bone regeneration ability of different forms of bovine bone scaffolds. The amount of rhBMP-2 has not been established or determined, and various concentrations were used for the purpose of this study. Our previous study showed that the BV of the particle bone with 50 µg rhBMP-2 was higher than that with 5 µg rhBMP-2 [36]. In another in vivo study, various amounts of 5, 10, and 30 µg rhBMP-2 were used for the animal experiment with different types of bone scaffold [44,45,46]. The group with 5 µg rhBMP-2 with dentin matrix scaffold showed 74.7% new bone formation, and the autogenous bone graft group showed 48% new bone formation in histomorphometric analysis after grafting in the alveolar bone defect of a beagle model [46]. When the rhBMP-2 with hyaluronic acid hydrogel scaffold was grafted onto the subperiosteal space of a rat calvarial bone, the BVs of the 30 µg rhBMP-2 with gel, 1 µg rhBMP-2 with gel, and only hydrogel groups were 139, 57, and 18 mm^3^, respectively [45]. Although the method of the experiment and the type of scaffold were different in each study, the rhBMP-2 application group showed more new bone formation than the group with bone scaffold graft alone. Further evaluation would be required to compare new bone formation with varying concentrations of rhBMP-2. 

In our study, rhBMP-2 significantly increased new bone formation using particle and block bovine bones, especially when used with block bone scaffolds. In addition to osteoinductive activity, rhBMP-2 also induces adipogenesis, leading to adipose tissue along with new bone formation. There was a significant difference in the BV between bone scaffolds, and increased ATV was observed in the B+BMP group; rhBMP-2 showed a high adipogenic and osteoinductive ability when used with block bovine bone and seemed to have better efficacy as a BMP carrier. In conclusion, rhBMP-2 could be effectively used with block bone scaffolds and showed excellent new bone formation. Further studies are required to evaluate the changes in adipose tissue inside new bone.

## 4. Materials and Methods

### 4.1. Experimental Animals and Study Design

Twenty-four 8-week-old Sprague Dawley rats (Samtako Biokorea, Osan, Korea) with an average weight of 250 g (200–300 g) were used in this study. The rats were housed two per cage under specific pathogen-free conditions with ad libitum access to a standard rodent diet and water. The animals were acclimated to the new environment for 14 days before surgery. This study was approved by the Institutional Animal Care and Use Committee of Jeonbuk National University Hospital, Jeonju, Republic of Korea (JBUH-IACUC-2020-17-1, 5 August 2020).

The rats were randomly divided into four groups (*n* = 6), and a subperiosteal pocket was created on the parietal bone of each rat calvarium. rhBMP-2 (COWELL^®^ BMP; Cowellmedi, Busan, Korea), demineralized bovine bone particle, and block (Bio-Oss^®^ small granules (0.25–1.0 mm) and Bio-Oss Collagen^®^, Geistlich Pharma AG, Wolhusen, Switzerland) were used in this study. rhBMP-2 was diluted with 0.3 mL of saline, and 30 µg doses of rhBMP-2 were prepared. In the particle (*n* = 6) and block (*n* = 6) groups, 0.08 g of bovine bone particle and a piece of the block were grafted onto the subperiosteal calvarial pocket, and the same weight of bovine bone particle and block with 30 µg of rhBMP-2 was grafted onto the same space in the P (particle)+BMP and B (block)+BMP groups. 

### 4.2. Surgical Procedure

General anesthesia was administered by intramuscular injection of a combination of Zoletil 50 (15 mg/kg; Vibac, Carros, France) and Rumpun (0.2 mL/kg; Bayer Korea, Seoul, Korea). The calvarial bone of the skull was shaved and disinfected with povidone-iodine. Local anesthesia was administered to the subdermal tissue on the calvarial bone with an injection of 2% lidocaine with epinephrine (1:100,000). A horizontal step incision of approximately 5 mm was made on the posterior portion of the calvarial bone. Subperiosteal dissection was performed to expose the calvarial bone, and a subperiosteal pocket was made on the surface of the parietal bone of the calvarium, and the particle and block bone were grafted into the pocket. In the P+BMP and B+BMP groups, after the bovine bone was grafted, diluted rhBMP-2 (30 µg) was applied onto the bone material in the subperiosteal pocket. After grafting, the muscle and skin were closed with 3-0 Vicryl (Ethicon, Inc., Somerville, NJ, USA). Gentamycin (1 mg/kg; Kookje, Seoul, Korea) and pyrin (0.5 mL/kg; Green Cross Veterinary Products, Seoul, Korea) were intramuscularly injected three times daily for 3 days. Six rats from each group were euthanized 6 weeks after the surgery. Specimens were fixed in 10% formalin. μ-CT and histological analyses were performed to evaluate the formation of new bone.

### 4.3. FE-SEM Analysis of Bone Scaffolds

The morphologies of the particle and block bovine bone scaffolds were examined using a field emission electron scanning microscope (SUPRA 40VP, ZEISS, Oberkochen, Germany) at the Center for University-Wide Research Facilities (CURF) at Jeonbuk National University (Jeonju-si, Korea).

### 4.4. μ-CT Analysis

All calvarial samples were analyzed by μ-CT at the Center for University-Wide Research Facilities (CURF) at Jeonbuk National University (Jeonju-si, Korea). The samples were evaluated using a SkyScan 1076 (Bruker, Karlsruhe, Belgium) with a pixel size of 35 µm. The following parameters of the CT scanner were set: 100 kV voltage for the X-ray tube, 100 μA current for the X-ray source, and 190 ms of exposure time. The detector and X-ray source were rotated by 0.6° in 360° rotation steps. The scanned images were reconstructed using the NRecon (Bruker, Karlsruhe, Germany). The region of interest of each sample represented the grafted area on the surface of the calvarial bone of the rat, and it was reconstructed in three-dimensional images for the analysis of BV, BMD, TbTh, TbSp, and ATV using CTAn (Bruker, Karlsruhe, Belgium) software. The threshold for the analysis of new bone was set in the range of 70–255, and the threshold for the adipose tissue was 20–70.

### 4.5. Histological Evaluation

After the μ-CT analysis, the samples were decalcified in 5% nitric acid for 2 weeks and dehydrated in ethyl alcohol and xylene. The samples were separated on a midline sagittal suture and embedded in paraffin blocks to examine the sagittal plane surfaces of the samples. The paraffin blocks were sliced into sections and stained with hematoxylin and eosin. The sections showing the sagittal image of the calvarial bone and grafted bone scaffold were selected for histological analysis. Stained tissue slides were examined using an Olympus BX51 microscope (Olympus, Tokyo, Japan). Digital images of the selected sections were captured using a digital camera (DP-73; Olympus, Tokyo, Japan). 

### 4.6. Immunohistochemical Evaluation of Osteogenic Markers

IHC analysis was performed with histological sections to evaluate the expression of osteogenic markers, namely, BSP and osteocalcin. Anti-BSP (GTX12155; GeneTex, Inc., Irvine, CA, USA) and anti-osteocalcin (sc-365797; Santa Cruz Biotechnology, Inc., Dallas, TX, USA) antibodies were used as primary antibodies. The Dako REAL EnVision Detection System (Dako, Glostrup, Denmark) was used for immunohistochemical staining according to the manufacturer’s instructions. Counterstaining was performed using Mayer’s hematoxylin (Sigma-Aldrich, Missouri, MO, USA). Stained tissue slides were examined with an Olympus BX51 microscope (Olympus, Tokyo, Japan), and images were captured using a digital camera (DP-73; Olympus, Tokyo, Japan).

### 4.7. Statistical Analysis

One-way analysis of variance (ANOVA; Version 23, SPSS Inc., Chicago, IL, USA) was used to compare four independent groups. Bonferroni’s method was used for post-hoc testing. Differences were considered significant at *p* < 0.05.

## Figures and Tables

**Figure 1 ijms-22-11121-f001:**
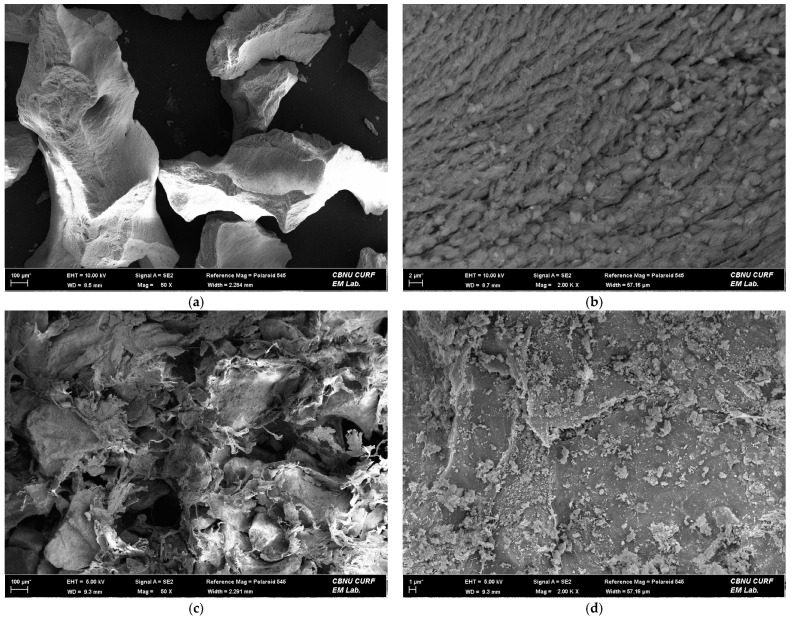
Field emission scanning electron microscope images of each bovine bone scaffold: Particle bovine bone scaffold ((**a**,**b**); original magnification 50×, 2000×, respectively), Block bovine bone scaffold ((**c**,**d**); original magnification 50×, 2000×, respectively).

**Figure 2 ijms-22-11121-f002:**
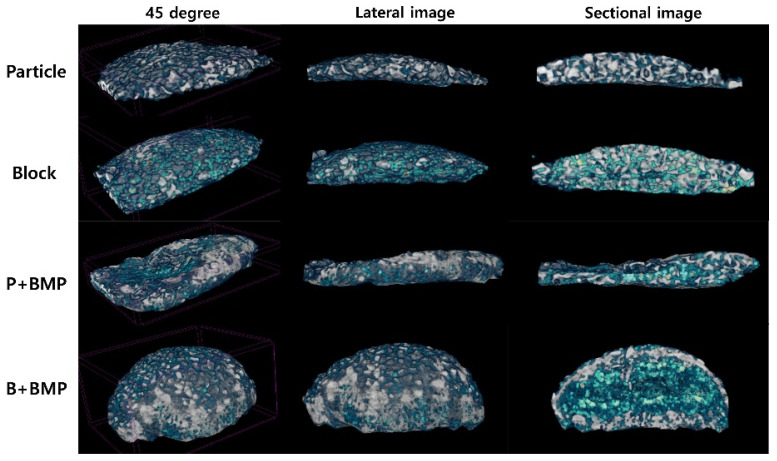
Three-dimensional reconstruction images observed with micro-computed tomography (μ-CT) analysis of particle (P), block (B), P+ bone morphogenetic protein (BMP), and B+BMP group showed the newly formed bone (green light) and grafted bone scaffold (white light).

**Figure 3 ijms-22-11121-f003:**
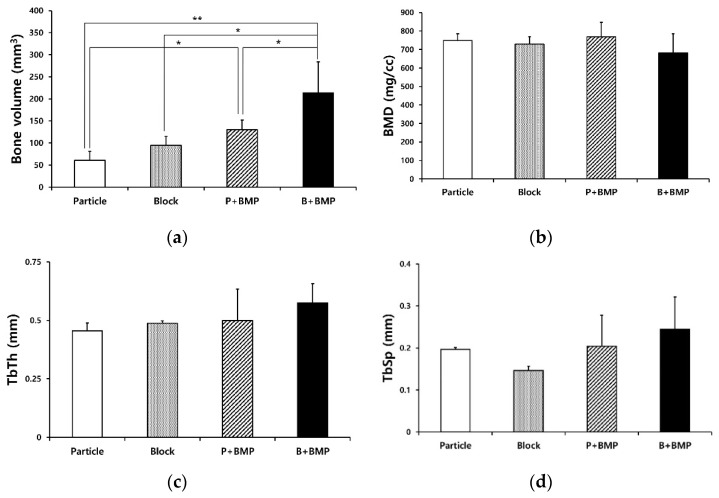
Micro-computed tomography analysis: (**a**) Bone volume (BV), (**b**) Bone mineral density (BMD), (**c**) Trabecular thickness (TbTh), and (**d**) Trabecular spacing (TbSp) of the the particle (P), block (B), P+ bone morphogenetic protein (BMP), and B+BMP groups. Significant differences were observed in the BV values (* *p* < 0.05, ** *p* < 0.001).

**Figure 4 ijms-22-11121-f004:**
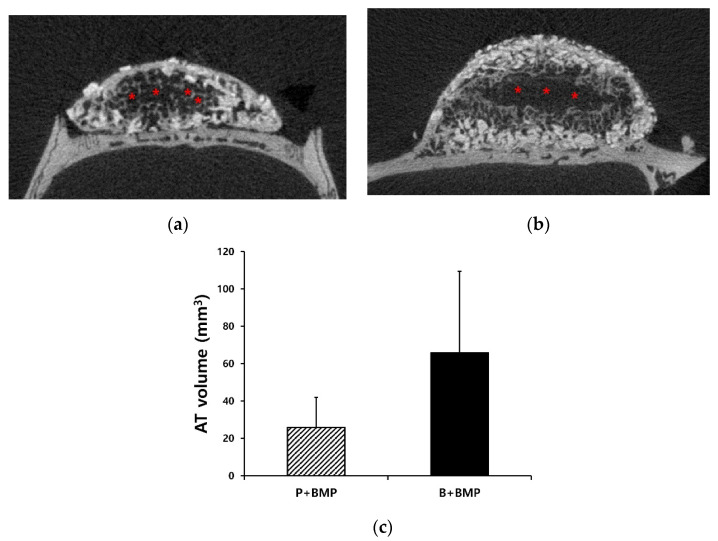
Micro-computed tomography images and volumes of the adipose tissue (AT) in the particle (P)+ bone morphogenetic protein (BMP) and block (B)+BMP groups. (**a**) Void space formation in the P+BMP group, (**b**) vacant space in the central area of the bone scaffold in the B+BMP group. (**c**) The average AT volumes of the P+BMP and B+BMP groups were 25.87 ± 16.03 and 65.81 ± 43.65 mm^3^ (*p* = 0.063). Red asterisks = adipose tissue.

**Figure 5 ijms-22-11121-f005:**
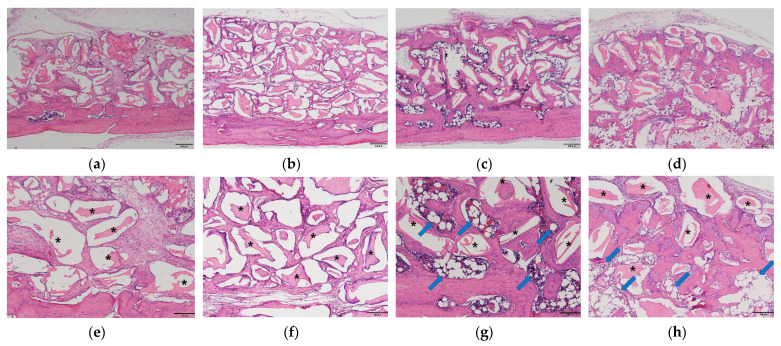
Histologic images (hematoxylin and eosin staining) of each group. (**a**,**e**) Particle (P), (**b**,**f**) Block (B), (**c**,**g**) P+bone morphogenetic protein (BMP), and (**d**,**h**) B+BMP groups. New bone formation and adipose tissue (blue arrow) were observed in the particle and B+BMP groups. (**e**–**h**) show high magnification images of (**a**–**d**), respectively. Black asterisks = bone scaffold ((**a**–**d**), original magnification 40×, bar = 200 µm; (**e**–**h**), original magnification 100×, bar = 100 µm).

**Figure 6 ijms-22-11121-f006:**
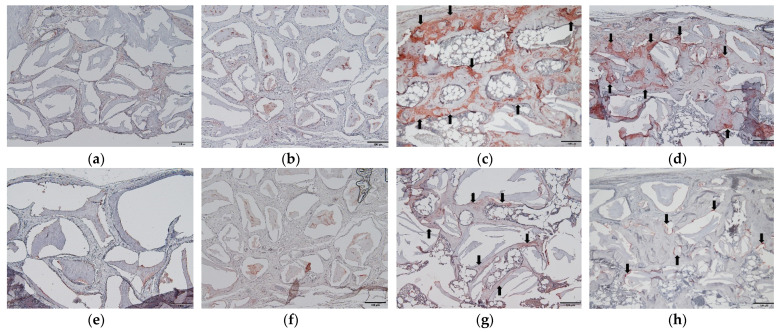
Immunohistochemical staining of bone sialoprotein (BSP) and osteocalcin observed in each group. (BSP; **a**–**d**) and osteocalcin (**e**–**h**) at 6 weeks after the operation. (**a**,**e**) particle (P), (**b**,**f**) block (B), (**c**,**g**) P+bone morphogenetic protein (BMP), and (**d**,**h**) B+BMP. A high expression of BSP and osteocalcin was observed in the new bone matrix of the P+BMP and B+BMP groups and is indicated by black arrows (Original magnification 100×, bar = 100 µm).

## Data Availability

Not applicable.

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
