# Peer review of "Comparison of Bone Regeneration in Different Forms of Bovine Bone Scaffolds with Recombinant Human Bone Morphogenetic Protein-2"

_ijms, 2021, doi:10.3390/ijms222011121_

Round 1
Reviewer 1 Report
In the manuscript entitled: “Comparison of bone regeneration in different forms of bovine bone scaffolds with recombinant human bone morphogenetic protein-2”, the authors compared the bone regeneration ability of particle and block bones, acting as bone scaffolds, with rhBMP-2, and evaluated them as rhBMP-2 carriers.
The authors found that The bone volume of the B+BMP group was significantly higher than that of the other groups (p < 0.00) with no significant difference in the bone mineral density. Adipose tissue formation was observed in the rhBMP-2 application group. The average adipose tissue volume of the B+BMP group was higher than that of the P+BMP group. Histologically, the particle and B+BMP groups showed a higher formation of the new bone. However, adipose tissue and void spaces were also formed, especially in the B+BMP group.
The authors concluded that rhBMP-2 could be effectively used with block bone scaffolds and showed excellent new bone formation.
Major comments:
In general, the idea and innovation of this study, regards analysis of bone scaffold is interesting, because the role of these factors in dentistry are validated but further studies on this topic could be an innovative issue in this field could be open a creative matter of debate in literature by adding new information. Moreover, there are few reports in the literature that studied this interesting topic with this kind of study design.
The study was well conducted by the authors; However, there are some concerns to revise that are described below.
The introduction section resumes the existing knowledge regarding the important factor linked with inflammatory mediators released during some oral diseases forms.
However, as the importance of the topic, the reviewer strongly recommends, before a further re-evaluation of the manuscript, to update the literature through read, discuss and must cites in the references with great attention all of those recent interesting articles, that helps the authors to better introduce and discuss the role of related biomarkers involved in OPG/RANKL such as Galectin, NLPR3 and transglutaminases: 1) Isola G, Polizzi A, Santonocito S, Alibrandi A, Williams RC. Periodontitis activates the NLRP3 inflammasome in serum and saliva. J Periodontol. 2021 May 19. doi: 10.1002/JPER.21-0049. 2) Isola G, Lo Giudice A, Polizzi A, Alibrandi A, Murabito P, Indelicato F. Identification of the different salivary Interleukin-6 profiles in patients with periodontitis: A cross-sectional study. Arch Oral Biol. 2021 Feb;122:104997. doi: 10.1016/j.archoralbio.2020.104997. 3) Isola G, Polizzi A, Alibrandi A, Williams RC, Lo Giudice A. Analysis of galectin-3 levels as a source of coronary heart disease risk during periodontitis. J Periodontal Res. 2021 Jun;56(3):597-605.
The authors should be better specified, at the end of the introduction section, the rational of the study and the aim of the study.
In the material and methods section, should better clarify Morphometric bone parameters measurements. Moreover, please more specify the scientists involved in the different stages of the study.
The discussion section appears well organized with the relevant paper that support the conclusions, even if the authors should better discuss the relationship between Galcectin-3, NLRP3, transglutaminases and inflammation during periodontitis. The conclusion should reinforce in light of the discussions.
In conclusion, I am sure that the authors are fine clinicians who achieve very nice results with their adopted protocol. However, this study, in my view does not in its current form satisfy a very high scientific requirement for publication in this journal and requests a revision before a futher re-evaluation of the manuscript.
Minor Comments:
Abstract:
- Better formulate the abstract section by better describing the aim of the study
Introduction:
- Please refer to major comments
Discussion
- Please add a specific sentence that clarifies the results obtained in the first part of the discussion
Author Response
Major comments:
In general, the idea and innovation of this study, regards analysis of bone scaffold is interesting, because the role of these factors in dentistry are validated but further studies on this topic could be an innovative issue in this field could be open a creative matter of debate in literature by adding new information. Moreover, there are few reports in the literature that studied this interesting topic with this kind of study design.
The study was well conducted by the authors; However, there are some concerns to revise that are described below.
The introduction section resumes the existing knowledge regarding the important factor linked with inflammatory mediators released during some oral diseases forms.
However, as the importance of the topic, the reviewer strongly recommends, before a further re-evaluation of the manuscript, to update the literature through read, discuss and must cites in the references with great attention all of those recent interesting articles, that helps the authors to better introduce and discuss the role of related biomarkers involved in OPG/RANKL such as Galectin, NLPR3 and transglutaminases: 1) Isola G, Polizzi A, Santonocito S, Alibrandi A, Williams RC. Periodontitis activates the NLRP3 inflammasome in serum and saliva. J Periodontol. 2021 May 19. doi: 10.1002/JPER.21-0049. 2) Isola G, Lo Giudice A, Polizzi A, Alibrandi A, Murabito P, Indelicato F. Identification of the different salivary Interleukin-6 profiles in patients with periodontitis: A cross-sectional study. Arch Oral Biol. 2021 Feb;122:104997. doi: 10.1016/j.archoralbio.2020.104997. 3) Isola G, Polizzi A, Alibrandi A, Williams RC, Lo Giudice A. Analysis of galectin-3 levels as a source of coronary heart disease risk during periodontitis. J Periodontal Res. 2021 Jun;56(3):597-605.
Thanks for your kind recommendation. I am really sorry to say one more time... The issue and topic of our article was related of the inflammatory biomarker and periodontitis. Our study was related with bone regeneration and rhBMP-2 as osteoinductive protein
The authors should be better specified, at the end of the introduction section, the rational of the study and the aim of the study.
Thanks for your kind advice. We revise the end of introduction and the aim of the study. And we refer to the English editing service one more time after extensive revision.
In the material and methods section, should better clarify Morphometric bone parameters measurements. Moreover, please more specify the scientists involved in the different stages of the study.
Thanks for your kind advice. As we said in previous review, We revise and specify our co-author in the section of author contribution.
In material and methods section, we added the info about the threshold of new bone and adipose tissue and analyzing software for measuring of bone parameters.
And we performed the English editing service one more time after the revision.
The discussion section appears well organized with the relevant paper that support the conclusions, even if the authors should better discuss the relationship between Galcectin-3, NLRP3, transglutaminases and inflammation during periodontitis. The conclusion should reinforce in light of the discussions.
Thanks for your kind advice. I really sorry to said that the issue and topic of our study did not have a lot of correlation with the inflammatory biomarker and periodontitis. Our study was for the bone regeneration and rhBMP-2 as osteoinductive protein. Sorry…
In conclusion, I am sure that the authors are fine clinicians who achieve very nice results with their adopted protocol. However, this study, in my view does not in its current form satisfy a very high scientific requirement for publication in this journal and requests a revision before a futher re-evaluation of the manuscript.
Minor Comments:
Abstract:
Better formulate the abstract section by better describing the aim of the study
Thanks for your kind advice. We revise the abstract section.
Reviewer 2 Report
I have previously reviewed the manuscript. The authors introduced some enhancements in the text. Still, there are some misleading statements that must be corrected. Again, the authors misinterpret the data, ignoring statistical analysis, which leads to unfounded conclusions.
Correct the following statements:
In Abstract: “The average adipose tissue volume of the B+BMP group was higher than that of the P+BMP group.”
In Results:
“The average TbTh of the B+BMP group was higher than that of the other groups. No significant difference was observed in the TbTh among the four groups (p > 0.05).”
“The average TbSp of the B+BMP group was higher than that of the other groups. No significant differences were observed between the values among all groups (p > 0.05).”
in Figure 4c description:
“Average AT volume of B+BMP group was higher than that of P+BMP group (p = 0.063).”
If the p > 0.05, the differences between groups are not significant, therefore it cannot be stated that some study groups present higher values of the parameter than others.
Discussion:
Rephrase the statements:
“In our study, the block bone with osteoconductive activity is used with osteoinductive growth factors such as rhBMP-2 to enhance the efficacy of bone regeneration and is a suitable scaffold as a BMP carrier.”
Correct:
“The BV of the particle group was 60.42 ± 20.35 mm3 and that of the block group was 94.46 ± 20.05 mm3. The average BV of block group was higher that of particle group, but there was no significant difference.”
Again, stick to the data. If p > 0.05, no significant differences BV between block and particle group were observed, hence it cannot be stated that BV of block group was higher that of particle group.
“In the previous our study, the 50 µg of rhBMP-2 was used with bovine particle bone and it showed the comparable bone regeneration. The B+BMP group was used with the 50 µg of rhBMP-2 and it showed more BV compared with the group of particle bone with the 50 µg of rhBMP-2 [34].”
Comparable to what? Please, correct.
Again, please compare the outcomes of the present study with findings from the literature.
“In the other in vivo study, varied amounts of 5, 10, 30, and 50 µg of rhBMP-2 have been used for the animal experiment with different kinds of bone scaffold [42-44].”
When authors mentioned that other studies used different amounts of rhBMP-2, please briefly add the outcomes.
Again, briefly add info on most desirable ranges of values of BV, TbTh, TbSp, and ATV parameters of newly formed bone or add data describing the bone obtained from the control group (where no bone scaffold was grafted).
Again, do higher values of all these parameters (BV, TbTh, TbSp, ATV) guarantee overall good bone quality and density?
Repetition of the statement: “In previous our study, the 50 µg of rhBMP-2 was used and it showed the comparable bone regeneration combined with particle bone scaffold [34].”
In Conclusions, please correct: “There was a significant difference in the BV and ATV between bone scaffolds;..” Stick to the data. No significant differences in ATV were observed.
Materials and Methods:
Again, please add in the text the explanation, how the following measurements of BV, BMD, TbTh, TbSp, and ATV were taken.
Correct grammar throughout the manuscript, esp. of newly added sentences.
Author Response
Correct the following statements:
In Abstract: “The average adipose tissue volume of the B+BMP group was higher than that of the P+BMP group.”
Thanks for your kind advice. We added the “although the difference was not significant”. We performed the English editing service one more time after the revision.
In Results:
“The average TbTh of the B+BMP group was higher than that of the other groups. No significant difference was observed in the TbTh among the four groups (p > 0.05).”
“The average TbSp of the B+BMP group was higher than that of the other groups. No significant differences were observed between the values among all groups (p > 0.05).”
Thanks for your kind advice. We revise that sentence.
in Figure 4c description:
“Average AT volume of B+BMP group was higher than that of P+BMP group (p = 0.063).”
If the p > 0.05, the differences between groups are not significant, therefore it cannot be stated that some study groups present higher values of the parameter than others.
Thanks for your kind advice. As your previous review comment, we added the level of significance. In the result, the average of the ATV was higher in B+BMP group and it is true.. is it problem? We just state the average was higher… and as your comment, we revise that sentence.
Discussion:
Rephrase the statements:
“In our study, the block bone with osteoconductive activity is used with osteoinductive growth factors such as rhBMP-2 to enhance the efficacy of bone regeneration and is a suitable scaffold as a BMP carrier.”
Thanks for your kind advice. We revised that sentence as “In our study, the block bone with osteoconductive activity was used with osteoinductive growth factors such as rhBMP-2 to enhance the efficacy of bone regeneration and the block bone was evaluated as a BMP carrier.” We performed the English editing service one more time after the revision.
Correct:
“The BV of the particle group was 60.42 ± 20.35 mm3 and that of the block group was 94.46 ± 20.05 mm3. The average BV of block group was higher that of particle group, but there was no significant difference.”
Again, stick to the data. If p > 0.05, no significant differences BV between block and particle group were observed, hence it cannot be stated that BV of block group was higher that of particle group.
Thanks for your kind advice. We just stated the average of BV was higher as the result. But, as your comments, we delete that sentence.
“In the previous our study, the 50 µg of rhBMP-2 was used with bovine particle bone and it showed the comparable bone regeneration. The B+BMP group was used with the 50 µg of rhBMP-2 and it showed more BV compared with the group of particle bone with the 50 µg of rhBMP-2 [34].”
Comparable to what? Please, correct.
Thanks for your kind advice. The term of “comparable” was used in the previous report [34]. We revise them.
Again, please compare the outcomes of the present study with findings from the literature.
Thanks for your kind advice. We add the outcomes of present study in the 6th paragraph. We performed the English editing service one more time after the revision.
“In the other in vivo study, varied amounts of 5, 10, 30, and 50 µg of rhBMP-2 have been used for the animal experiment with different kinds of bone scaffold [42-44].”
When authors mentioned that other studies used different amounts of rhBMP-2, please briefly add the outcomes.
Thanks for your kind advice. We add the outcomes of other studies. We performed the English editing service one more time after the revision.
Again, briefly add info on most desirable ranges of values of BV, TbTh, TbSp, and ATV parameters of newly formed bone or add data describing the bone obtained from the control group (where no bone scaffold was grafted).
I am really sorry to say this reply. The desirable or ideal value of new bone formation parameter can be differed by experiment and it would be impossible to define and decided the reference value of the new bone formation. There has been no report that said the desirable range of micro CT result. Please check it one more time. The recent publication of IJMS “Int. J. Mol. Sci. 2021, 22(15), 8101”, “Int. J. Mol. Sci. 2021, 22(16), 9089” did not provide the desirable range of values of micro CT result.
I will say one more time, our experiment design did not make defect on the calvarial bone, and just graft on the surface of calvarial bone and this model was useful and wide used in the onlay graft experiment. There were previously reported articles “Tissue Eng Regen Med 2016;13(3):311-321”, “Int J Oral Maxillofac Surg 2010; 39; 883-888”. In those study, the control group was the group that was grafted only the scaffold like our study.
Again, do higher values of all these parameters (BV, TbTh, TbSp, ATV) guarantee overall good bone quality and density?
Thanks for your kind advice. The higher values of BV and TbTh was considered new bone formation and new bone regeneration. you can find it recent study “Int. J. Mol. Sci. 2021, 22(15), 8101”, “Int. J. Mol. Sci. 2021, 22(16), 9089”.
Repetition of the statement: “In previous our study, the 50 µg of rhBMP-2 was used and it showed the comparable bone regeneration combined with particle bone scaffold [34].”
Thanks for your kind advice. We delete that sentence.
In Conclusions, please correct: “There was a significant difference in the BV and ATV between bone scaffolds;..” Stick to the data. No significant differences in ATV were observed.
Thanks for your kind advice. We revise that sentence.
Materials and Methods:
Again, please add in the text the explanation, how the following measurements of BV, BMD, TbTh, TbSp, and ATV were taken.
Thanks for your kind advice. In previous your review, we added the information in the material section. We designed the region of interest and set up the threshold for each parameter and then the software of CTAn analyzed the parameter. Everything was described in the material. Please, there is nothing more to explain. We performed the English editing service one more time after the revision.
Correct grammar throughout the manuscript, esp. of newly added sentences.
Thanks for your kind advice. We refer to the English editing service one more time and then resubmit. Thanks regard.
Round 2
Reviewer 1 Report
The authors have revised almost of reviewer's comments.
Author Response
The authors have revised almost of reviewer's comments.
Thanks for your kind review of our manuscript.
Reviewer 2 Report
Thank you for providing the revised manuscript and implementing the changes.
One minor issue:
Please introduce the authors’ reply to my comment:
“Again, do higher values of all these parameters (BV, TbTh, TbSp, ATV) guarantee overall good bone quality and density?”
Authors’ reply: “Thanks for your kind advice. The higher values of BV and TbTh was considered new bone formation and new bone regeneration. you can find it recent study “Int. J. Mol. Sci. 2021, 22(15), 8101”, “Int. J. Mol. Sci. 2021, 22(16), 9089”.
into the manuscript (Discussion section preferably).
Thank you
Author Response
Thanks for your kind review of our manuscript. We introduce our reply and your comment in the discussion section.
This manuscript is a resubmission of an earlier submission. The following is a list of the peer review reports and author responses from that submission.
Round 1
Reviewer 1 Report
In this manuscript, Seok and colleagues compared the efficacy of particle and block bovine bone scaffolds with recombinant human bone morphogenetic protein-2 for rat calvarial bone regeneration in rat. Overall, the study is straightforward and finding can potentially provide guidance to bone regeneration in clinical settings. However, the reviewer felt the manuscript may not be good fit for the journal International Journal of Molecular Sciences. A tissue engineering journal would fit the focus of this manuscript better. Please find my major as well as minor comments below:
Major comments:
- The overall experiment design is suffering from lacking a defect group where no scaffold or BMP is applied.
- It is interesting that both BMP treated groups have large void area, the authors assumed it is filled with adipocytes and attributed it to adipogenic effect of BMP. However, it is unclear if the B or P alone groups also have this void area. Additionally, from the histology images, the adipocytes are scattered around the remaining scaffolds and the newly formed bones, which does not seem to be consistent with the large continuous void area seen on CT.
- Since either the P or the B scaffold alone does not have much effect on the bone regeneration, one may wonder would be sufficient to just add BMP for bone regeneration?
Minor comments:
- Why the 3D reconstruction images of the defect area are so different sized among groups? Would this suggest variations during surgery? If so, this would affect data interpretation.
- Not sure why some texts are in red.
- It is unknown from the method, how the authors combine BMP and the scaffolds or they were actually applied to the defects separately.
Author Response
Major comments:
1.The overall experiment design is suffering from lacking a defect group where no scaffold or BMP is applied.
Thanks for your kind advice. Our experiment design was the bone grafting on the subperiosteal space on the calvarial bone. There was no bone defect and the graft material was just grafting on the calvarial bone and the graft material was on the space between cortical space of calvarial bone and periosteum. This model was known and reported as only graft or subperiosteal bone graft. This model showed the bone regeneration ability of the graft material supposed to the vertical bone augmentation and only bone graft. And those model has been used in the field of dental implantology research. There was no defect group where no scaffold or BMP is applied. The control group with no bone graft is just the animal itself. Those group was not required for the evaluation and that was the overuse of animals.
2.It is interesting that both BMP treated groups have large void area, the authors assumed it is filled with adipocytes and attributed it to adipogenic effect of BMP. However, it is unclear if the B or P alone groups also have this void area. Additionally, from the histology images, the adipocytes are scattered around the remaining scaffolds and the newly formed bones, which does not seem to be consistent with the large continuous void area seen on CT.
Thanks for your kind advice. In figure 5 g and h, you could see the formation of adipocyte both BMP treated group. But in the figure 5 e and f, there was no adipocyte or void area around grafted material or newly formed bone. The vacant area was observed around the grafted material but it did not the adipocyte or adipose tissue.
3.Since either the P or the B scaffold alone does not have much effect on the bone regeneration, one may wonder would be sufficient to just add BMP for bone regeneration?
Thanks for your kind advice. As you know, bone regeneration can be formed by the process of the osteogenesis, osteoinduction, and osteoconduction. The bone scaffold has only the ability of osteoconduction not provide the osteoinduction and osteogenesis ability. Osteogenesis can be performed by the transplantation of the osteogenetic cell and tissue. Osteoinduction is formation of new bone by the stimulation of the mesenchymal cell by the bone-inductive protein such as bone morphogenetic proteins. Currently, the rhBMP-2 has been widely used for the bone grafting operation for the dental implant installation. Transplantation of the autologous osteogenetic cell and tissue would be best to new bone formation but it can lead to donor site morbidity. So currently, the bone grafting various biocompatible bone scaffold and rhBMP-2 is considered favorable option for bone grafting in dental implantology.
Minor comments:
1.Why the 3D reconstruction images of the defect area are so different sized among groups? Would this suggest variations during surgery? If so, this would affect data interpretation.
Thanks for your kind advice. We grafted same amount of bone graft material in the subperiosteal pocket on the calvarial bone not in the defect. So the morphology in each group was a little different and the particle type bone also could affect the morphology of newly formed bone. The figure 2 was the crop image of the grafted bone and did not present the calvarial bone. We surely extracted the grafted bone using the NRecon software and analyzed that region for the measurement.
2.Not sure why some texts are in red.
Thanks for your kind advice. This manuscript was firstly reviewed in the special issue of biomaterial for bone tissue engineering and we received the major revision from first reviewers. This manuscript was first revised according to the first major revision we add the SEM analysis and revised some sentences. And during review process, we encouraged from the IJMS editorial office to make this manuscript resubmit and it may be because of the close deadline of the special issue.
3.It is unknown from the method, how the authors combine BMP and the scaffolds or they were actually applied to the defects separately.
Thanks for your kind advice. The rhBMP-2 can be used in diluted with saline. We made the subperiosteal pocket on the calvarial bone of rat and grafted same weight of bone scaffold whether particle and block and then put on the rhBMP-2 diluted with saline on the grafted scaffold in the subperiosteal pocket with using pipette. This application was not a very difficult operation and all experiment was done well.
Reviewer 2 Report
In the manuscript entitled: “Comparison of bone regeneration in different forms of bovine bone scaffolds with recombinant human bone morphogenetic protein-2”, the authors aimed to examine the bone regeneration ability of particle and block bones, acting as bone scaffolds, with rhBMP-2, and evaluated them as rhBMP-2 carriers.
The authors found that The bone volume of the block/BMP group was significantly higher than that of the other groups. The trabecular thickness of Block/BMP showed marginally increases compared to that of the particle/BMP group with no significant difference in the bone mineral density. Adipose tissue formation was observed in the rhBMP-2 application group. The adipose tissue volume of the block/BMP group showed marginal increases compared to that of the particle/BMP group. Histologically, the particle and block/BMP groups showed a higher formation of the new bone.
The authors concluded that the prevalence of CHD significantly increases (p<0.05) with the aging of patients and a higher functional class of heart failure (p<0.05). The results obtained demonstrate that CHD is interrelated with periodontal diseases.
Major comments:
In general, the idea and innovation of this study, regards analysis of the interaction between bmp in alveolar bone is interesting, because the role of these factors in dentistry are validated but further studies on this topic could be an innovative issue in this field could be open a creative matter of debate in literature by adding new information. Moreover, there are few reports in the literature that studied this interesting topic with this kind of study design.
The study was well conducted by the authors; However, there are some concerns to revise that are described below.
The introduction section resumes the existing knowledge regarding the important factor linked with inflammatory mediators and growth factors associated with periodontal ligament.
However, as the importance of the topic, the reviewer strongly recommends, before a further re-evaluation of the manuscript, to update the literature through read, discuss and must cites in the references with great attention all of those recent interesting articles, that helps the authors to better introduce and discuss the role of bmp and related inflammatory biomarkers as Il-1B and transglutaminases: 1) Isola G, Lo Giudice A, Polizzi A, Alibrandi A, Murabito P, Indelicato F. Identification of the different salivary Interleukin-6 profiles in patients with periodontitis: A cross-sectional study. Arch Oral Biol. 2021 Feb;122:104997. doi: 10.1016/j.archoralbio.2020.104997. 2) Currò M, Matarese G, Isola G, Caccamo D, Ventura VP, Cornelius C, Lentini M, Cordasco G, Ientile R. Differential expression of transglutaminase genes in patients with chronic periodontitis. Oral Dis. 2014 Sep;20(6):616-23. doi: 10.1111/odi.12180.
The authors should be better specified, at the end of the introduction section, the rational of the study and the aim of the study. In the material and methods section, should clarify the adipose tissue formation in the particle/BMP. Moreover, please more specify the scientists involved in the different stages of the study.
The discussion section appears well organized with the relevant paper that support the conclusions, even if the authors should better discuss the relationship between NLRP3, IL-1b, transglutaminases, BMP and inflammation. The conclusion should reinforce in light of the discussions.
In conclusion, I am sure that the authors are fine clinicians who achieve very nice results with their adopted protocol. However, this study, in my view does not in its current form satisfy a very high scientific requirement for publication in this journal and requests a revision before a futher re-evaluation of the manuscript.
Minor Comments:
Abstract:
- Better formulate the abstract section by better describing the aim of the study
Introduction:
- Please refer to major comments
Discussion
- Please add a specific sentence that clarifies the results obtained in the first part of the discussion
- Page 7 last paragraph: Please reorganize this paragraph that is not clear
Author Response
Major comments:
In general, the idea and innovation of this study, regards analysis of the interaction between bmp in alveolar bone is interesting, because the role of these factors in dentistry are validated but further studies on this topic could be an innovative issue in this field could be open a creative matter of debate in literature by adding new information. Moreover, there are few reports in the literature that studied this interesting topic with this kind of study design.
The study was well conducted by the authors; However, there are some concerns to revise that are described below.
The introduction section resumes the existing knowledge regarding the important factor linked with inflammatory mediators and growth factors associated with periodontal ligament.
However, as the importance of the topic, the reviewer strongly recommends, before a further re-evaluation of the manuscript, to update the literature through read, discuss and must cites in the references with great attention all of those recent interesting articles, that helps the authors to better introduce and discuss the role of bmp and related inflammatory biomarkers as Il-1B and transglutaminases: 1) Isola G, Lo Giudice A, Polizzi A, Alibrandi A, Murabito P, Indelicato F. Identification of the different salivary Interleukin-6 profiles in patients with periodontitis: A cross-sectional study. Arch Oral Biol. 2021 Feb;122:104997. doi: 10.1016/j.archoralbio.2020.104997. 2) Currò M, Matarese G, Isola G, Caccamo D, Ventura VP, Cornelius C, Lentini M, Cordasco G, Ientile R. Differential expression of transglutaminase genes in patients with chronic periodontitis. Oral Dis. 2014 Sep;20(6):616-23. doi: 10.1111/odi.12180.
Thanks for your kind recommendation. I said to sorry that issue and topic of those article that is the related of the inflammatory biomarker and periodontitis does not have a lot of correlation with our study that is related with bone regeneration and rhBMP-2 as osteoinductive protein.
The authors should be better specified, at the end of the introduction section, the rational of the study and the aim of the study. In the material and methods section, should clarify the adipose tissue formation in the particle/BMP. Moreover, please more specify the scientists involved in the different stages of the study.
Thanks for your kind comment. As you said, the methodology of the μCT analysis was important and the method of analysis should be exact, specific, and reproducible. The special algorithm with thresholding is critical to distinguish and measure the specific area of target bone. In this study, we seat up the threshold with reference to the histological images in each group. The threshold of adipose tissue was 20 to 70. For the readers, the specific and detail method would be good, but the setting of threshold can be varied by each experiments and studies. The setting value of threshold can be differed and effected by the μCT instrument, software, animal species, and, bone scaffold material. So this value can be varied each experiment. So we were questionable to describe the specific threshold setting value for reader that can be varied. We concerned that can be lead to confusing and misunderstanding to readers because it can be different in each laboratory. We revise the end of introduction and the aim of the study. We revise and specify our co-author in the section of author contribution.
The discussion section appears well organized with the relevant paper that support the conclusions, even if the authors should better discuss the relationship between NLRP3, IL-1b, transglutaminases, BMP and inflammation. The conclusion should reinforce in light of the discussions.
Thanks for your kind recommendation. I said to sorry that issue and topic of those article that is the related of the inflammatory biomarker and periodontitis does not have a lot of correlation with our study that is related with bone regeneration and rhBMP-2 as osteoinductive protein.
In conclusion, I am sure that the authors are fine clinicians who achieve very nice results with their adopted protocol. However, this study, in my view does not in its current form satisfy a very high scientific requirement for publication in this journal and requests a revision before a futher re-evaluation of the manuscript.
Thanks for your kind comment. As you said, our study design was more benefit and useful in clinician and it would look did not had high scientific priority. But this clinical results are very worth to the implant dentistry and I am sure that this study was helpful to the both of clinician and scientist. In this point of view, our study goes well and satisfy to be published in the special issue of “Biomaterial for Bone Tissue Engineering 2.0”. Thanks regard.
Minor Comments:
Abstract:
Better formulate the abstract section by better describing the aim of the study
Thanks for your kind comment. Thanks for your kind comment. We stated the aim of this study on the first sentence of the abstract. We revise the sentence.
Introduction:
Please refer to major comments
Discussion
Please add a specific sentence that clarifies the results obtained in the first part of the discussion
Page 7 last paragraph: Please reorganize this paragraph that is not clear
Thanks for your kind comment. We delete some sentences and revise the last paragraph in page 7. We state our result of study in the first part of discussion.
Reviewer 3 Report
The paper “Comparison of bone regeneration in different forms of bovine bone scaffolds with recombinant human bone morphogenetic protein-2” compared the use of particle and block bovine bone scaffolds as carriers for rh-BP-2 protein and their effectiveness in bone regeneration.
The manuscript is quite well written, yet some issues must be addressed:
Abstract:
Line 15: please, reorder the statement “Demineralized bovine bone particles, blocks, and rhBMP-2 were grafted into the subperiosteal space of rat calvarial bone, and the rats were divided into the particle, block, particle/BMP, and block/BMP groups.” as it seems that first, the rats were divided into groups, and then the bone scaffolds were grafted.
The aim of the study should be clearly stated. Define “efficacy of block bone scaffolds as rhBMP-2 carriers”.
Results:
Please, unify the names of the study groups, in the text and on Figures. Use: particle/BMP and block/BMP, instead of P+BMP and B+BMP, respectively.
Statements in lines 118-121 could be shortened and significant differences could be presented in charts in Figure 3 clearer.
Lines 129-130: as the difference is not statistically significant, omit this statement. Correct the chart in Figure 3c.
Figures 3 and 4 are of low resolution, replace them with sharper image, correct the description (remove value p<0.1) as only at p-values of less than 0.05 the differences were considered significant.
Please, remove statements like: “There was a marginally significant difference between the values of the two groups (p = 0.063).” as marginally significant difference means simply that there was no significant difference. As stated in Methods Section, the authors considered differences significant at p-values of less than 0.05.
Revise the description of results and Discussion, accordingly
Discussion section seems bit chaotic and could be shortened.
Line 241: revise statement: if there was no significant difference in bone volume between the block and particle bone, then on what ground the authors concluded that “(…) block scaffold showed a higher bone formation and was a more favorable scaffold in the subperiosteal onlay bone graft than the particle bone.”?
Line 247: repetition of statement from Result section
Briefly add info on most desirable values of BV, TbTh, TbSp, and ATV parameters of newly formed bone.
Line 290: add reference to this statement. Shorten this paragraph (290-299).
Line 324: repetition of information, remove.
Line 305: explain briefly the role of BSP and osteocalcin
Compare the outcomes of the present study with findings from the literature.
Line 328: revise to: “Apart from the osteoinductive activity, rhBMP-2 also induces adipogenesis, leading to adipose tissue formation and new bone formation.”
Add limitations of the study.
Add conclusions, or rather make them clearer (lines 334-337).
Materials and Methods:
Please add the explanation, how the following measurements of BV, BMD, TbTh, TbSp, and ATV were taken.
Author Response
Abstract:
Line 15: please, reorder the statement “Demineralized bovine bone particles, blocks, and rhBMP-2 were grafted into the subperiosteal space of rat calvarial bone, and the rats were divided into the particle, block, particle/BMP, and block/BMP groups.” as it seems that first, the rats were divided into groups, and then the bone scaffolds were grafted.
Thanks your kind comment. We revise that sentences.
The aim of the study should be clearly stated. Define “efficacy of block bone scaffolds as rhBMP-2 carriers”.
Thanks for your kind comment. We stated the aim of this study on the first sentence of the abstract. We revise the aim of this study and could not clearly understand of your comments of the definition of the “efficacy of block bone scaffold as rhBMP-2 carriers”. This study was to compare and confirm the best combination of the rhBMP-2 and bone scaffold for the new bone regeneration.
Results:
Please, unify the names of the study groups, in the text and on Figures. Use: particle/BMP and block/BMP, instead of P+BMP and B+BMP, respectively.
Thanks your kind comment. We revise all of it in the manuscript, figures, and figure legends.
Statements in lines 118-121 could be shortened and significant differences could be presented in charts in Figure 3 clearer.
Thanks your kind comment. We thought that the exact P value between each groups should be described. The significant differences were clearly showed in the figure 3 but there was not stated the exact P value and so we stated the exact P value in the lines 118-121 of result.
Lines 129-130: as the difference is not statistically significant, omit this statement. Correct the chart in Figure 3c.
Thanks for your kind comment. We removed the result of marginally significant difference.
Figures 3 and 4 are of low resolution, replace them with sharper image, correct the description (remove value p<0.1) as only at p-values of less than 0.05 the differences were considered significant.
Thanks for your kind comment. We removed the result of marginally significant difference and revised the figure of chart.
Please, remove statements like: “There was a marginally significant difference between the values of the two groups (p = 0.063).” as marginally significant difference means simply that there was no significant difference. As stated in Methods Section, the authors considered differences significant at p-values of less than 0.05.
Thanks for your kind comment. We removed the result of marginally significant difference.
Revise the description of results and Discussion, accordingly
Discussion section seems bit chaotic and could be shortened.
Line 241: revise statement: if there was no significant difference in bone volume between the block and particle bone, then on what ground the authors concluded that “(…) block scaffold showed a higher bone formation and was a more favorable scaffold in the subperiosteal onlay bone graft than the particle bone.”?
Thanks for your kind comment. We just state the result of BV in the particle and block group, that was 60.42 and 94.46 in each group. As your comment, we delete the “and was a more favorable scaffold” in this sentence.
Line 247: repetition of statement from Result section
Thanks for your kind comment. We revise and delete that sentence.
Briefly add info on most desirable values of BV, TbTh, TbSp, and ATV parameters of newly formed bone.
Thanks for kind comment. The desirable value of each parameter are useful information for the evaluation of the bone formation. However, the desirable or ideal value of new bone formation parameter can be differed by experiment and it would be impossible to define and decided the reference value of the new bone formation. Because the qualitative and quantitative value of new bone formation was affected by various factors of amount of graft, kind of graft material, duration of bone regeneration, and species of animal. So it can be different result by experiment. So like most experiment, we compared the bone regeneration ability of the P+BMP and B+BMP groups comparing with control group. And we could find via this experiment block bone scaffold with rhBMP-2 was ideal combination for the new bone formation.
Line 290: add reference to this statement. Shorten this paragraph (290-299).
Thanks for your kind comment. The paragraph (290-299) was from the same reference [39]. We make this paragraph shorten.
Line 324: repetition of information, remove.
Thanks for your kind comment. We remove that sentence.
Line 305: explain briefly the role of BSP and osteocalcin
Thanks for your kind comment. We add the brief explanation of BSP and osteocalcin in discussion
Compare the outcomes of the present study with findings from the literature.
Line 328: revise to: “Apart from the osteoinductive activity, rhBMP-2 also induces adipogenesis, leading to adipose tissue formation and new bone formation.”
Thanks for your kind comment. We revise it.
Add limitations of the study.
Thanks for your kind comment. We already described the limitation of our study in the paragraph (308-315). We performed this experiment only use of 30 µg of rhBMP-2. So we said the further study that compare the new bone formation applied varied amounts of 5, 10, 30, and 50 µg of rhBMP-2 with different kinds of bone scaffold would be required.
Add conclusions, or rather make them clearer (lines 334-337).
Thanks for your kind comment. We revise the conclusion shorter ad clearer.
Materials and Methods:
Please add the explanation, how the following measurements of BV, BMD, TbTh, TbSp, and ATV were taken.
Thanks for your kind advice. As you said, the methodology of the μCT analysis was important and the method of analysis should be exact, specific, and reproducible. The special algorithm with thresholding is critical to distinguish and measure the specific area of target bone. In this study, we seat up the threshold with reference to the histological images in each group. The threshold of total bone was 70 to 255, that of new bone was 90-160, and that of adipose tissue was 20 to 70. For the readers, the specific and detail method would be good, but the setting of threshold can be varied by each experiments and studies. The setting value of threshold can be differed and effected by the μCT instrument, software, animal species, and, bone scaffold material. So this value can be varied each experiment. So we were questionable to describe the specific threshold setting value for reader that can be varied. We concerned that can be lead to confusing and misunderstanding to readers because it can be different in each laboratory.
We already describe the software commercial name as “The scanned images were reconstructed using the NRecon (Bruker, Germany)”. And the measurements of BV, BMD, TbTh, TbSp, and ATV was analyzed by this software.
Round 2
Reviewer 2 Report
Unfortunately, the real limits and bias of the original version of the manuscript were not solved.
The overall experiment design is still suffering from lacking a defect group where no scaffold or BMP it was applied. Moreover, no reason to add BMP for bone regeneration in different critical defects
Reviewer 3 Report
The following issues must be addressed:
Introduction:
Rephrase the statement:
“Therefore, in this study, we aimed to compare bone regeneration ability in the different forms of bovine bone scaffolds with rhBMP-2 and evaluate the efficacy of block bone scaffolds as rhBMP-2 carriers.”
First, it should be “efficacy of different forms of bone scaffolds as rhBMP-2 carriers”.
Second, define, what makes specific form of a bone scaffold an efficient rhBMP-2 carrier.
Results:
Rephrase the statement:
“The average TbTh of the B+BMP group was higher than that of the other groups, and the average trabecular spacing (TbSp) values of the particle, block, P+BMP, and B+BMP groups were 0.20, 0.15 ± 0.01, 0.20 ± 0.07, and 0.24 ± 0.08 mm, respectively (Figure 3D).”
- Do not mix description of results of different measurements in one sentence. Split this sentence in two.
- Was the average TbTh of the B+BMP group significantly higher (?!) than that of the other groups? If not, add this info in the sentence
Remove “marginally significant” from Figure 3 description, as statistically speaking TbTh values were not different between the study groups.
Figure 4c please add the level of significance in the figure and in Figure description. Also, add in the sentence in introduction
“The average adipose tissue volume of the B+BMP group was higher than that of the P+BMP group”
whether it was significantly higher or not.
Figures 3 and 4 are of low resolution, replace them with sharper image.
Discussion:
Rephrase the statement:
“In our study, the block bone with osteoconductive activity is used with osteoinductive growth factors such as rhBMP-2 to enhance the efficacy of bone regeneration and is a suitable scaffold as a BMP carrier.”
These two statements:
“The BV of the particle group was 60.42 ± 20.35 mm3 and that of the block group was 94.46 ± 20.05 mm3. Although there was no significant difference, the block bone scaffold showed a higher bone formation than the particle bone in the subperiosteal onlay bone graft.”
are bit contradictory: if there was no significant difference in bone volume between the block and particle bone, then on what ground the authors concluded that “(…) block scaffold showed a higher bone formation”? Please, rephrase these statements. Stick to the data.
Again, please compare the outcomes of the present study with findings from the literature.
When authors mentioned that other studies used different amounts of rhBMP-2, please briefly add the outcomes.
Briefly add info on most desirable ranges of values of BV, TbTh, TbSp, and ATV parameters of newly formed bone or add data describing the bone obtained from the control group (where no bone scaffold was grafted).
Do higher values of all these parameters (BV, TbTh, TbSp, ATV) guarantee overall good bone quality and density?
Add limitations of the study, aside from the amount of rhBMP-2 used in the experiment, e.g. limited observation time etc.
Add Conclusions section, shorten conclusions, make them clear and concise (now there are too many repetitions of info).
Materials and Methods:
Again, please add in the text the explanation, how the following measurements of BV, BMD, TbTh, TbSp, and ATV were taken.
Correct grammar throughout the manuscript, esp. of newly added sentences. Use past tense when describing your experiment and its outcomes.